# Enhanced Serum IgG Detection Potential Using 38KD-MPT32-MPT64, CFP10-Mtb81-EspC Fusion Protein and Lipoarabinomannan (LAM) for Human Tuberculosis

**DOI:** 10.3390/pathogens11121545

**Published:** 2022-12-15

**Authors:** Zhuohong Yan, Xiaojue Wang, Ling Yi, Bin Yang, Panjian Wei, Hongyun Ruan, Jinghui Wang, Xinting Yang, Hongtao Zhang

**Affiliations:** 1Department of Central Laboratory, Beijing Chest Hospital, Capital Medical University, Beijing Tuberculosis and Thoracic Tumor Research Institute, Beijing 101149, China; 2Department of Medical Oncology, Beijing Chest Hospital, Capital Medical University, Beijing Tuberculosis and Thoracic Tumor Research Institute, Beijing 101149, China; 3The Third Department of Tuberculosis, Beijing Chest Hospital, Capital Medical University, Beijing 101149, China

**Keywords:** tuberculosis, IgG, fusion protein, 38KD-MPT32-MPT64, CFP10-Mtb81-EspC, Ag85B-HBHA, lipoarabinomannan (LAM)

## Abstract

For the rapid, reliable, and cost-effective methods of tuberculosis (TB) auxiliary diagnosis, antibody (Ab) detection to multiple antigens of *Mycobacterium tuberculosis* (Mtb) has great potential; however, this methodology requires optimization. We constructed 38KD-MPT32-MPT64, CFP10-Mtb81-EspC, and Ag85B-HBHA fusion proteins and evaluated the serum Ab response to these fusion proteins and to lipoarabinomannan (LAM) by ELISA in 50 TB patients and 17 non-TB subjects. IgG responses to the three fusion proteins and to LAM were significantly higher in TB patients, especially in Xpert Mtb-positive TB patients (TB-Xpert^+^), than in non-TB subjects. Only the anti-38KD-MPT32-MPT64 Ab showed higher levels in the Xpert Mtb-negative TB patients (TB-Xpert^−^) than in the non-TB, and only the anti-LAM Ab showed higher levels in the TB-Xpert^+^ group than in the TB-Xpert^−^ group. Anti-Ag85B-HBHA Ab-positive samples could be accurately identified using 38KD-MPT32-MPT64. The combination of 38KD-MPT32-MPT64, CFP10-Mtb81-EspC, and LAM conferred definite complementarity for the serum IgG detection of TB, with relatively high sensitivity (74.0%) and specificity (88.2%). These data suggest that the combination of 38KD-MPT32-MPT64, CFP10-Mtb81-EspC, and LAM antigens provided a basis for IgG detection and for evaluation of the humoral immune response in patients with TB.

## 1. Introduction

Tuberculosis (TB) was the leading cause of mortality from a single infectious microorganism for the 4 years [1] predating the severe acute respiratory syndrome coronavirus 2 (SARS-CoV-2) pandemic [2,3]. *Mycobacterium tuberculosis* (Mtb) is the pathogenic causative agent of TB. Pulmonary tuberculosis (PTB) constitutes approximately 80% of the total cases. The diagnosis of TB is challenging because there is no single, optimal method for its diagnosis. As the gold standard of TB diagnosis, bacteriological culture tests are time-consuming. Acid fast staining of sputum smears has low sensitivity and requires a high bacillary content (5000–10,000 CFU/mL) in the sputum [4]. Xpert MTB/RIF, a nucleic acid amplification-based assay, is expensive and requires specialized equipment.

Immunological methods, such as the tuberculin skin test and the interferon gamma release assay, are primarily used for screening Mtb infection but cannot discriminate active TB and latent TB infection (LTBI). New diagnostic technologies, especially rapid detection technologies, therefore still need to be developed. Lipoarabinomannan (LAM) is cell wall lipoglycan specific to the mycobacterium species and is a biomarker for TB. However, point-of-care (POC) methods to detect LAM in urine using immunoassays have poor sensitivity and are limited to only HIV-coinfected TB diagnosis [5,6,7]. Recently, global TB LAM, in-vivo and urinary LAM structure have been elucidated [7,8], which suggested approaches to develop specific antibodies (Abs) for POC tests for LAM in the urine of suspected TB patients and may allow for improvements in diagnosis performance.

Abs are candidate biomarkers of TB. The Ab responses to Mtb-specific antigens in patients with TB are heterogenous. A number of Mtb antigens have been explored for auxiliary TB diagnosis [9]. The use of a single antigen has failed to achieve reproducible sensitivity and specificity concurrently [10,11]. We previously found that the 38KD (antigen 5, Rv0934), MPT32 (*M. tuberculosis* protein 32, Rv1860), MPT64 (Rv1980c), EspC (ESX-1 substrate protein C, Rv3615c), Mtb81 (Rv1837c), and lipoarabinomannan (LAM) antigens had relatively high frequencies of IgG detection in TB patients, and showed high specificity (90–96.7%) [10]. These antigens showed some complementarity for IgG detection. Ag85B (Rv1886c) is a diagnostic and/or vaccine antigen [12]. Heparin-binding hemagglutinin adhesion (HBHA; Rv0475) protein is a mycobacterial cell surface protein, and a promising biomarker in TB [13]. Ag85B-HBHA (a nano-AH vaccine) can induce robust humoral immune responses [14]. Fusion proteins contain multiple epitopes of the individual antigens, which have the potential to increase the sensitivity of Ab detection [11,15].

On the other hand, humoral immunity is a major arm of the adaptive immune system but its role in TB has remained largely undetermined. A series of independent studies has shown that Abs may contribute significantly to reducing the mycobacterial burden [16]. A post-hoc analysis of the MVA85A vaccine trial found that elevated Ag85A-specific IgG levels correlated with a reduced risk of TB development [17], and there is increasing evidence that Abs confer protection against Mtb [16,18]. Antibodies from individuals with LTBI have unique IgG crystallizable fragment (Fc) effector profiles and distinct IgG Fc glycosylation [18,19,20,21], as well as selective binding to the low affinity immunoglobulin γ Fc region receptor IIIA (FcγRIIIA) found on innate immune cells [18,19], such as natural killer cells and macrophages, which are actively recruited and may contribute to antimicrobial activity [20]. BCG-mediated humoral immunity has been reported to be heterogeneous [22]. Intradermal (ID) BCG immunization could introduce Mtb-specific Abs, the level of which increased significantly with an increase in the dose and immunization time; however, some studies have reported the opposite results [22,23]. In this regard, Ab detection could be used to evaluate the humoral immune status of patients with TB.

In the current study, we determined the sensitivity and specificity of 38KD-MPT32-MPT64, CFP10-Mtb81-EspC, and Ag85B-HBHA fusion proteins, and LAM polysaccharide antigen, alone and in combination, for auxiliary diagnosis of TB and optimized the combined detection using the 38KD-MPT32-MPT64 and CFP10-Mtb81-EspC fusion proteins and LAM.

## 2. Materials and Methods

### 2.1. Population Study and Sample Collection

This study enrolled 50 patients with active pulmonary TB (PTB) and 17 non-TB subjects from the Beijing Chest Hospital. PTB was diagnosed based on the results of sputum smears and culture tests, Xpert MTB/RIF assays, radiology (chest X-ray or computed tomography), interferon gamma release assays, and clinical signs and symptoms. The diagnostic criteria for TB were according to WS 288-2017 Tuberculosis Diagnosis published in 2017 [24]. Of the 50 PTB patients, 38 patients had not received TB-specific treatment, and 12 patients were within one week of TB-specific treatment initiation upon enrollment. Furthermore, 37 PTB patients had received BCG vaccination, 2 patients had not received BCG vaccination, and 11 patients had an unknown BCG vaccination status. No patients were administered immunosuppressive drugs.

The PTB patients were categorized into several subgroups. Based on culture results, they were divided into the culture-positive group (n = 37) and culture-negative group (n = 13). According to their Xpert results, they were divided into the Xpert MTB-positive group (TB-Xpert^+^, n = 42) and Xpert MTB-negative group (TB-Xpert^−^, n = 8). Based on the concurrence of type 2 diabetes mellitus (DM), patients were divided into the non-diabetic TB group (TB-Non-DM, n = 38) and the diabetic TB group (TB-DM, n = 12). Patients in the TB-DM group all had a history of type 2 DM and were undergoing regular DM treatment. Based on initial anti-TB drug treatment, the patients were divided into the non-treatment TB group (TB-Non-T, n = 38) and the treatment TB group (TB-T, n = 12). Non-TB showed no clinical signs or symptoms of TB, and had no history of either TB infection or DM. Non-TB had received BCG vaccination. All participants were HIV-negative. Serum samples were collected, aliquoted, and stored at −80 °C until use.

### 2.2. Antigen Preparation

The recombinant fusion proteins 38KD-MPT32-MPT64 (Rv0934-Rv1860-Rv1980c), CFP10-Mtb81-EspC (Rv3874-Rv1837c-Rv3615c), and Ag85B-HBHA (Rv1886c-Rv0475) were engineered by inserting a linker with eight residues (GGGSGGGS) between the adjacent antigens (i.e., 38KD-Linker-MPT32-Linker-MPT64, CFP10-Linker-Mtb81-Linker-EspC, and Ag85B-Linker-HBHA). The coding sequences of the three fusion proteins were synthesized by Sangon Biotech (Shanghai, China) with specific endonuclease restriction sites (*Nde*I/*Xho*I for 38KD-MPT32-MPT64; *Bam*HI/*Xho*I for CFP10-Mtb81-EspC and Ag85B-HBHA) and cloned into expression vectors with His-tag (LEHHHHHH) or/and T7 tag (MASMTGGQQMGRGS) incorporation (His tag for 38KD-MPT32-MPT64; T7 and His tag for CFP10-Mtb81-EspC and Ag85B-HBHA), resulting in the generation of pET-30a-(38KD-MPT32-MPT64), pET-21a-(CFP10-Mtb81-EspC), and pET-21a-(Ag85B-HBHA). The recombinant plasmids were transformed into Rosetta (DE3) to express the fusion proteins. All fusion proteins were induced via treatment with 0.5 mM isopropyl β-D-1-thiogalactopyranoside overnight with shaking at 220 rpm at 20 °C, and were then purified by metal-chelate column chromatography using Ni-NTA resin (Qiagen, Valencia, CA, USA). The resulting protein concentrations were determined using a non-interference protein assay kit (Sangon Biotech). The molecular weights and purities of the fusion proteins were estimated via sodium dodecyl sulfate-polyacrylamide gel electrophoresis (SDS-PAGE), followed by staining with Coomassie brilliant blue. Individual preparations of proteins with a purity of at least 90% were further aliquoted, then stored at −80 °C.

### 2.3. ELISA

Human serum IgG against the 38KD, MPT32, MPT64, CFP10, Mtb81, EspC, 38KD-MPT32-MPT64, CFP10-Mtb81-EspC, and Ag85B-HBHA fusion proteins and LAM polysaccharide antigen was detected by ELISA. Purified H37Rv LAM (NR-14848) was obtained from BEI Resources (National Institute of Allergy and Infectious Diseases, National Institutes of Health, Bethesda, MD, USA). These assays were performed using the commercial BD OptEIA™ Reagent Set B ELISA kit (BD Biosciences, San Diego, CA, USA). Microplate wells were coated with 5 µg/mL fusion protein (overnight at 4 °C) or 1 µg/mL LAM (overnight at 37 °C) in coating buffer (0.1 M sodium carbonate, pH 9.5), blocked with 5% skim milk in wash buffer and then incubated overnight at 4 °C with serum samples diluted 1:50 (for detecting Abs against the fusion proteins) or 1:200 (for detecting Abs against LAM) in 5% skim milk. After five washes, the plates were incubated with horseradish peroxidase-coupled goat anti-human IgG (H + L) Ab at a 1:60,000 dilution (Sigma-Aldrich, St. Louis, MO, USA) for 1 h at 37 °C. A final incubation with TMB substrate solution for 30 min at 37 °C was used for Ab-antigen detection. After the addition of stop solution (BD Biosciences), the optical density at 450 nm (OD_450_) was measured.

### 2.4. Statistical Analysis

Statistical analysis was conducted using GraphPad Prism 7 software (GraphPad Software Inc., San Diego, CA, USA) and SPSS25. Statistical analysis was performed using a Mann–Whitney test. ROC curves were used to compare the efficacy of Mtb-specific IgG detection in patients with TB. Cut-off values were either determined using an ELISA based on data from non-TB and defined as the mean OD_450_ value + 2 standard deviations (SD) or were determined from ROC curves as the optimal cut-off values when the Youden Index (sensitivity + specificity − 1) was maximal. The positive predictive value (PPV), negative predictive value (NPV), and accuracy (ACC) were also analyzed. The efficacy for the combination of three fusion proteins and LAM was analyzed using a statistical method. The predicted probability for IgG detection with the combination of antigens was calculated using binary logistic regression, and the ROC curve was drawn with the predicted probability.

## 3. Results

### 3.1. Clinical and Demographic Characteristics of the Study Subjects

A total of 67 participants were included in this study, including 50 PTB patients and 17 non-TB subjects. The clinical and demographic characteristics of the individual study members are summarized in Table 1. Of the 50 patients with PTB, 74.0% (n = 37) were male, 26.0% (n = 13) were female; the mean age was 45.42 ± 18.59 years, 74.0% (n = 37) and 26.0% (n = 13) of patients showed positive (culture^+^) and negative (culture^˗^) mycobacterial culture results, respectively; 84.0% (n = 42) and 16.0% (n = 8) of patients showed positive (TB-Xpert^+^) and negative (TB-Xper^˗^) Xpert MTB/RIF results, respectively; 76% (n = 38) and 24% (n = 12) of patients were divided into TB-Non-DM (n = 38) and TB-DM (n = 12), respectively; and 76% and 24% of patients were divided into the non-treatment TB and treatment TB groups. Additionally, 100% of patients had chest X-ray results that were suggestive of PTB. Of the 17 non-TB subjects, 76.5% (n = 13) were male, 23.5% (n = 4) were female, and the mean age was 39.82 ± 5.60 years. All non-TB subjects had received BCG vaccination.

### 3.2. Generation of Fusion Antigens

In this study, we constructed and expressed three fusion proteins, including 38KD-MPT32-MPT64, CFP10-Mtb81-EspC, and Ag85B-HBHA, which contained 867, 981, and 514 amino acids (aa), with predicted molecular weights of 89, 105, and 55 kDa, respectively. The molecular weights of the three fusion proteins observed via SDS-PAGE were consistent with the predicted values, and fusion protein preparations with a purity of >90% (Figure 1) were used in all relevant experiments in this study.

### 3.3. Increased IgG Detection Ability Using 38KD-MPT32-MPT64 and CFP10-Mtb81-EspC Fusion Proteins Compared with Individual Antigens in Patients with PTB

We compared the IgG detection efficiency of 38KD-MPT32-MPT64 and CFP10-Mtb81-EspC fusion proteins with each individual antigen using samples from patients with TB (n = 50) and non-TB (n = 17) by ELISA (Figure 2A,B and Table 2). We found that the IgG levels to 38KD-MPT32-MPT64 and CFP10-Mtb81-EspC fusion proteins, and their individual antigens, were significantly increased in TB patients compared with the non-TB, with the exception of EspC (Figure 2A,B). ROC curves were used to compare the IgG detection ability of fusion proteins with each individual antigen. The area under the ROC curves (AUCs) for 38KD, MPT32, MPT64, and 38KD-MPT32-MPT64 were 0.7788 (95% CI, 0.6619–0.8957), 0.7729 (95% CI, 0.6560–0.8899), 0.7006 (95% CI, 0.5560–0.8452), and 0.8588 (95% CI, 0.7672–0.9504), respectively (Figure 2C). The AUCs for CFP10, Mtb81, EspC, and CFP10-Mtb81-EspC were 0.6747 (95% CI, 0.5330–0.8164), 0.6900 (95% CI, 0.5551–0.8249), 0.5853 (95% CI, 0.4494–0.7211), and 0.7118 (95% CI, 0.5878–0.8357), respectively (Figure 2D). The results indicated that fusion proteins had improved AUC values compared with individual antigens.

To determine the efficacy of fusion proteins on the antigenicity of individual antigens, we constructed matrix plots to compare the IgG detection results for fusion proteins and individual antigens for each patient with PTB. A sample was identified as being positive for TB-specific Ab when cut-off values were set as the mean OD_450_ + 2SD of samples from the non-TB. The results indicated that 36.0% (18/50), 36.0% (18/50), 24% (12/50), and 54% (27/50) of the PTB population elicited an IgG reaction to 38KD, MPT32, MPT64, and 38KD-MPT32-MPT64, respectively, and the specificity of the 38KD-MPT32-MPT64 was 94.1% (16/17) (Figure 2A). Furthermore, for 94.4% (17/18), 88.9% (16/18), and 83.3% (10/12) of the IgG-positive samples to 38KD, MPT32, and MPT64 antigens, respectively, an IgG reaction to 38KD-MPT32-MPT64 fusion protein was detected (Figure 2E).

In addition, 20% (10/50), 26% (13/50), 26% (13/50), and 40% (20/50) of the PTB population elicited an IgG Ab response to CFP10, Mtb81, EspC, and CFP10-Mtb81-EspC, respectively, and the specificity was between 94.1% (16/17) and 100% (17/17) (Figure 2B). Moreover, for 90% (9/10), 92.3% (12/13), and 61.5% (8/13) of the IgG-positive samples to 38KD, MPT32, and MPT64 antigens, respectively, an IgG reaction to 38KD-MPT32-MPT64 fusion protein was detected (Figure 2F). Taken together, these results indicated that the fusion proteins showed improved detection of IgG compared with individual antigens. We concluded that 38KD-MPT32-MPT64 and CFP10-Mtb81-EspC fusion proteins had increased sensitivity, and similar or slightly improved specificity in IgG detection compared with individual antigens.

### 3.4. IgG to Ag85B-HBHA Fusion Protein and LAM Polysaccharide Antigen in Patients with PTB

Ag85B is a secretory protein that is involved in the synthesis of cell wall mycolic acid in mycobacteria [25]. HBHA is a mycobacterial cell surface protein. LAM is a major constituent of the Mtb cell wall and is a mycobacterial outer surface antigen. The IgG levels to Ag85B-HBHA fusion protein and LAM antigen were significantly higher in the PTB group (n = 50) than in the non-TB group (n = 17) (Figure 3).

### 3.5. Characteristics of the IgG Reaction to Antigen Alone, and in Combination, among Patients with PTB

First, ROC curves were constructed to compare the efficacy of Mtb-specific IgG detection in patients with TB. The AUC values for 38KD-MPT32-MPT64, CFP10-Mtb81-EspC, Ag85B-HBHA, LAM, and the four-antigen combination were 0.8588 (95% CI, 0.7672–0.9504), 0.7118 (95% CI, 0.5878–0.8357), 0.7000 (95% CI, 0.5681–0.8319), 0.8294 (95% CI, 0.7082–0.9506), and 0.8965 (95% CI, 0.8163–0.9766) (Figure 4A,B). When the optimal cut-off values based on the ROC curve were applied, the sensitivities of the IgG detection to 38KD-MPT32-MPT64, CFP10-Mtb81-EspC, Ag85B-HBHA, LAM, and the four-antigen combination were 82.0% (95% CI, 68.56–91.42%), 50.0% (95% CI, 35.53–64.47%), 58.0% (95% CI, 43.21–71.81%), 82.0% (95% CI, 68.56–91.42%), and 84.0% (95% CI, 70.89–92.83%), with respective specificities of 82.4% (95% CI, 56.57–96.2%), 94.1% (95% CI, 71.31–99.85%), 82.4% (95% CI, 56.57–96.2%), 76.47% (95% CI, 50.1–93.19%), and 82.4% (95% CI, 56.57–96.2%).

When cut-off values were set as the mean OD_450_ + 2SD of the samples from the non-TB in ELISA, 54.0% (27/50), 40.0% (20/50), 30.0% (15/50), and 28.0% (14/50) of the PTB population elicited IgG to 38KD-MPT32-MPT64, CFP10-Mtb81-EspC, Ag85B-HBHA, and LAM, respectively, and the specificities of the tests by ELISA were 94.1% (16/17), 100.0% (17/17), 94.1% (16/17), and 94.1% (16/17) (Figure 2A,B, Figure 3, and Figure 4C, and Table 2). Using antigen combination analysis, 37 (74%) patients were recognized as TB-specific Ab-positive (TB-Ab^+^), and 13 (26%) patients were recognized as TB-specific Ab-negative (TB-Ab^˗^) (Figure 4C). We created a Venn diagram to show the recognition features of each type of antigen-related IgG in all of the TB˗Ab^+^ samples (Figure 5A).

Among the 37 TB˗Ab^+^ patients, the number of patients with IgG to any one, two, three, or four antigen(s) was respectively sixteen, seven, ten, and four, respectively. We found that the samples with elevated IgG to Ag85B-HBHA could be completely identified by their reaction to 38KD-MPT32-MPT64. We concluded that 38KD-MPT32-MPT64, CFP10-Mtb81-EspC, and LAM had clear complementarity for IgG detection. The sensitivity, specificity, PPV, NPV, and ACC using this combination for IgG detection were 74.0%, 88.2%, 94.9%, 53.6%, and 77.6%, respectively (Table 2). The combination of 38KD-MPT32-MPT64, CFP10-Mtb81-EspC, and LAM had the same AUC (0.8965) when compared with the four-antigen combination.

### 3.6. Characteristics of IgG to Antigens among Multiple Subgroups of TB Patients

Serum IgG to the four antigens in PTB patients divided into three of six subgroups (TB-Xpert^+^ or TB-Xpert^−^, TB-Non-DM or TB-DM, and TB-Non-T and TB-T) (Figure 5 and Table 3). The levels of anti-38KD-MPT32-MPT64 Ab were higher in the TB-Xpert^+^ and TB-Xpert^−^ groups than in the non-TB group, with no significant difference between the two TB groups; the levels of anti-CFP10-Mtb81-EspC and anti-Ag85B-HBHA Abs were both higher in the TB-Xpert^+^ group than in the non-TB group, with no statistical difference between the TB-Xpert^−^ and non-TB groups or between the TB-Xpert^+^ and TB-Xpert^−^ groups; and the levels of anti-LAM Ab were higher in the TB-Xpert^+^ group than in the TB-Xpert^−^ and non-TB groups, with no statistical difference between the TB-Xpert^−^ and non-TB groups (Figure 5A). The levels of IgG to the four antigens were significantly higher in culture^+^ TB group than in non-TB group, and the levels of IgG to 38KD-MPT32-MPT64, CFP10-Mtb81-EspC fusion protein, and LAM were significantly higher in culture^−^ TB group than in the non-TB group with the exception of Ag85B-HBHA, and no statistical difference between culture^+^ and culture^−^ TB groups was shown (data not shown here).

The levels of IgG to the four antigens were all significantly higher in the TB-Non-DM and TB-DM groups than in the non-TB, with no statistical difference in the IgG levels between the TB-Non-DM and TB-DM groups (Figure 5B), demonstrating that concurrent DM in TB patients does not affect the detection of Mtb-specific Abs.

The levels of IgG to 38KD-MPT32-MPT64 and LAM were significantly higher in the TB-Non-T and TB-T groups when compared with the non-TB group, and there was no statistical difference between the TB-Non-T and TB-T groups (Figure 5C). IgG levels to CFP10-Mtb81-EspC and Ag85B-HBHA were higher in the TB-Non-T group compared with the non-TB, with no statistical difference between the TB-Non-T and TB-T groups, or between the TB-T and non-TB groups. IgG levels to LAM tended to decrease in TB-T group compared with TB-Non-T group but did not reach statistical significance (*p* = 0.0541). The sensitivities of IgG detection in TB subgroups were summarized in Table 3.

## 4. Discussion

Numerous studies have reported the presence of multiple specific antibodies in patients with TB, which not only reflect the status of humor immunity but could also be used in the auxiliary diagnosis of TB. The application of a single antigen with good specificity often has low sensitivity in anti-Mtb Ab detection, whereas the combination of multiple antigens potentially reduces specificity. In our previous study, we screened a panel of Mtb-specific antigens with relatively high sensitivities and specificities in TB humoral immunity assessment, which included 38KD, MPT32, MPT64, Mtb81, EspC, and LAM (10). In this study, we constructed and expressed three fusion proteins to facilitate and optimize Mtb-specific Ab detection in PTB patients, including 38KD-MPT32-MPT64, CFP10-Mtb81-EspC, and Ag85B-HBHA fusion proteins.

We first evaluated the immunoreactivity of the fusion proteins and found that the 38KD-MPT32-MPT64 and CFP10-Mtb81-EspC fusion protein had higher antigenicity than individual antigens. The fusion protein could identify most of the samples identified by individual antigens. Moreover, the 38KD-MPT32-MPT64 and CFP10-Mtb81-EspC fusion protein had an increased IgG detection efficacy compared with the individual antigens. On combined analysis of the antigens (such as 38KD˗MPT32˗MPT64 vs 38KD+MPT32+MPT64), although the detection sensitivity using fusion proteins was slightly decreased, the specificity was higher. Some samples were identified as IgG positive using 38KD-MPT32-MPT64 or CFP10-Mtb81-EspC but were IgG negative using individual antigens. This finding may reflect the additive effects of individual antigen IgG levels that are undetectable in separate analyses. Over the course of our experiment, we found that 38KD-MPT32-MPT64 fusion protein was not suitable for lyophilization, because it was difficult to reconstitute after lyophilization.

The level of IgG to 38KD-MPT32-MPT64, CFP10-Mtb81-EspC, and Ag85B-HBHA fusion proteins were significantly increased in PTB patients compared with the non-TB. 38KD-MPT32-MPT64 fusion protein had the highest sensitivity. Importantly, the IgG to 38KD-MPT32-MPT64 was significantly increased in the TB-Xpert^−^ group compared with the non-TB group, which may be associated with their high immunogenic and secretory properties. CFP10-Mtb81-EspC fusion protein showed the highest specificity, which may reflect on the presence of antigens that are absent in the BCG strain and most non-TB mycobacterial species, such as CFP10 and EspC [26]. Anti-Ag85B-HBHA Ab-positive samples could be reliably identified using 38KD-MPT32-MPT64.

LAM is a lipoglycan that serves as a potent virulence factor that modulates the host immune response and plays an important role in the pathogenesis of Mtb infection [27,28]. Virulent Mtb and *M. bovis* species express mannose-capped LAM [29]. LAM-guided anti-TB treatment in HIV-TB patients seems associated with decreased mortality [30]. De et al. have recently determined the structures of LAM from three epidemiologically important lineages of Mtb, and identified a series of tailoring modifications that impact antibody binding [7,31]. In our previous study, we successfully generated anti-LAM monoclonal Abs using the spleen cells of a rabbit immunized with cell-wall components from Mtb *H37Rv*, which performed well at recognizing LAM from slow growing pathogenic mycobacteria, including Mtb *H37Rv*, *M. bovis*, and 96% (48/50) of the Mtb clinical isolates [32]. In this study, anti-LAM IgG Ab levels were significantly increased in the Xpert^+^ group compared with the Xpert^−^ group. Therefore, we speculate that the level of anti-LAM Ab may be closely related to the bacterial load. Anti-LAM Abs have been proven to play an important protective role in the host immune response to Mtb infection in animal models [33,34,35]. Anti-LAM Ab may interfere with Mtb virulence by preventing macrophage uptake through mannose receptors [35].

38KD, MPT32, MPT64, Ag85B, HBHA, and LAM antigens were also expressed by the BCG strain. However, IgG levels to these antigens were hardly detectable in BCG-vaccinated non-TB subjects, and the effectiveness of BCG in preventing disease is much lower in adults [36]. These findings may suggest a dramatic waning of the immune response after BCG vaccination in early childhood. Additionally, Mtb-specific IgG detection may be a potential assay to distinguish between LTBI and active TB. Antibody-based diagnostics for active TB and LTBI have focused on a small number of Mtb antigens, including LAM, purified protein derivatives, ESAT6/CFP10, Ag85A/B, and MPT64 [18,37]. In our study, the IgG to 38KD-MPT32-MPT64, CFP10-Mtb81-EspC, and LAM showed definite complementarity for the evaluation of humoral immunity in patients with active TB, and for the auxiliary diagnosis of TB. The combination of 38KD-MPT32-MPT64, CFP10-Mtb81-EspC, and LAM antigens showed a relatively high sensitivity (74.0%) and specificity (88.2%) in Mtb-specific IgG detection. Additionally, there were seven clinically diagnosed TB patients with negative culture and Xpert-MTB results in this study, and IgG reaction to 38KD-MPT32-MPT64, CFP10-Mtb81-EspC, and LAM could detect 71.4% (5/7) of these patients.

DM is a risk factor for the development of active TB. Insulin-resistant type 2 DM accounts for 90% of the global cases of DM and a large proportion of the people afflicted with the dual TB/DM burden [38]. In our study, 24% of the subjects with active TB concurrently suffered from type 2 DM. The IgG against the four antigens showed no significant differences between the TB-DM and TB-Non-DM groups. Anti-TB treatment may influence the IgG detection of some antigens, especially the cell wall associated antigens, such as LAM, however this needs to be further verified.

## 5. Conclusions

Immunoassays represent rapid detection technology and warrant further exploration in the context of TB prevention and control, including auxiliary diagnostic value and preliminary disease screening. Although we constructed 38KD-MPT32-MPT64, CFP10-Mtb81-EspC, and Ag85B-HBHA fusion proteins that showed increased IgG detection efficacy compared with the individual antigens, these finding constitute a pilot study and some limitations need to be taken into account, such as the limited sample size and the lack of consideration of latent TB infection and extrapulmonary TB. However, the preliminary results of this study support further in-depth validation, including testing various clinical patients and specimens.

## Figures and Tables

**Figure 1 pathogens-11-01545-f001:**
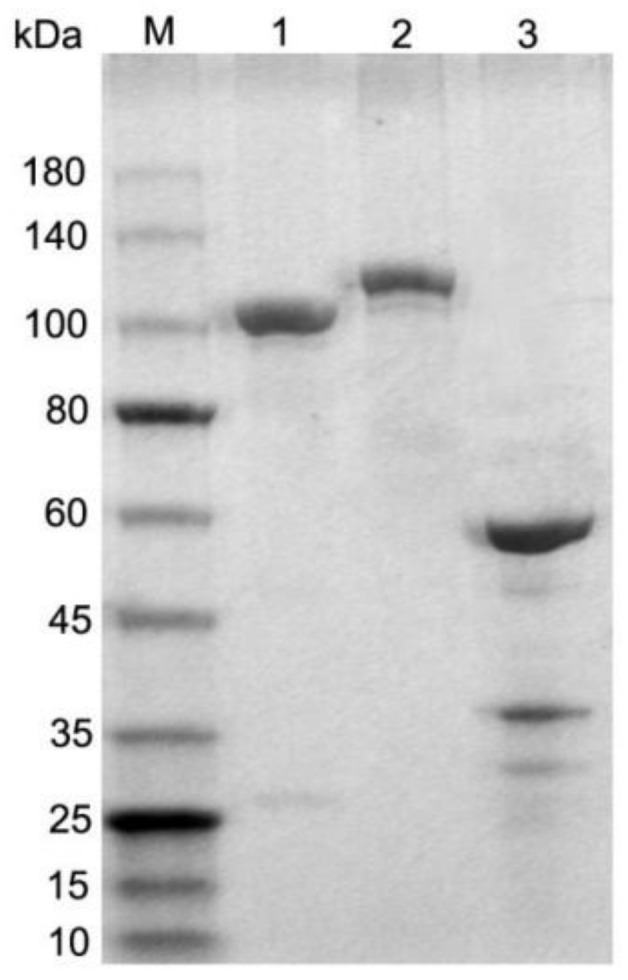
Analysis of purified tuberculosis fusion proteins by SDS-PAGE. M = molecular weight marker; 1 = 38KD-MPT32-MPT64; 2 = CFP10-Mtb81-EspC; 3 = Ag85B-HBHA. Molecular weights are shown in kDa.

**Figure 2 pathogens-11-01545-f002:**
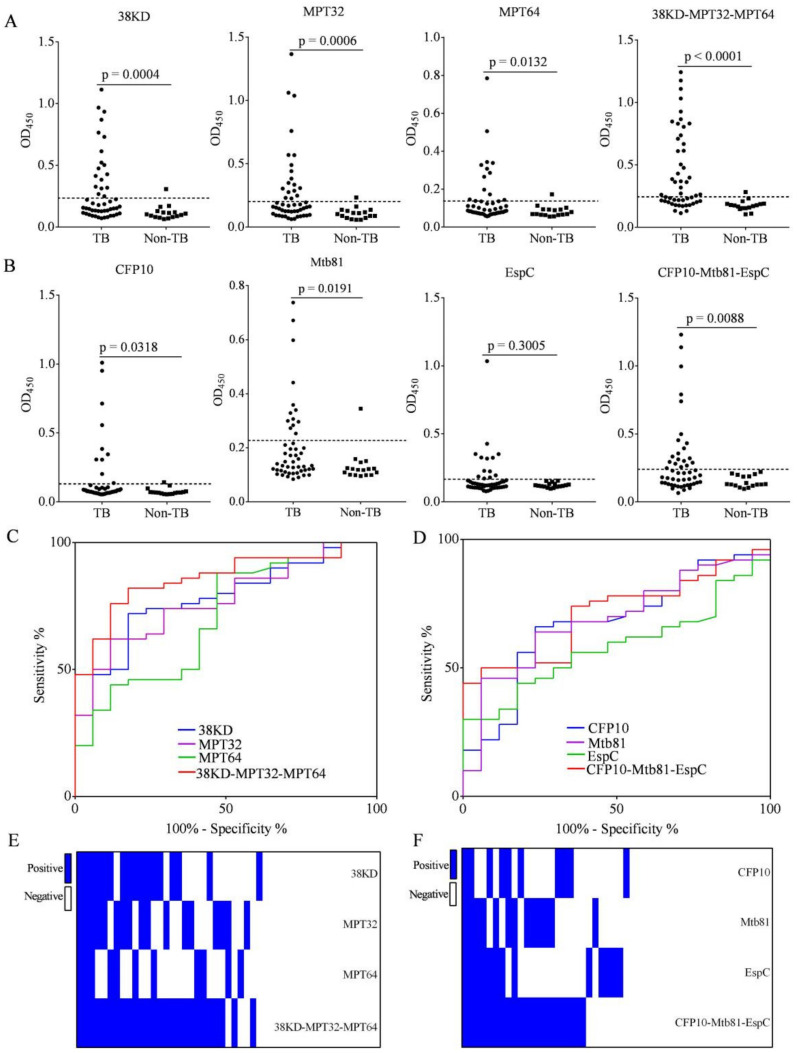
IgG detection by fusion proteins and individual Mtb-specific antigens in patients with tuberculosis (TB) and non-TB subjects. (**A**,**B**) Scatter plot comparing IgG levels detected by fusion proteins and individual Mtb-specific antigens using samples from patients with TB (n = 50) and non-TB (n = 17) by ELISA. The graphs show the OD_450_ values (from ELISA) for samples from all participants. Horizontal lines indicate the cut-off, based on the mean value from non-TB plus two standard deviations. (**C**,**D**) The receiver operating characteristic (ROC) curve analysis for evaluating the IgG detection capacity of fusion proteins compared with individual antigens. (**E**,**F**) Matrix plots comparing the IgG detection results between fusion proteins and individual antigens in each sample from patients with TB (n = 50). Mtb-specific IgG-positive and -negative samples are indicated by blue and white bars, respectively. Differences were assessed by a Mann–Whitney test.

**Figure 3 pathogens-11-01545-f003:**
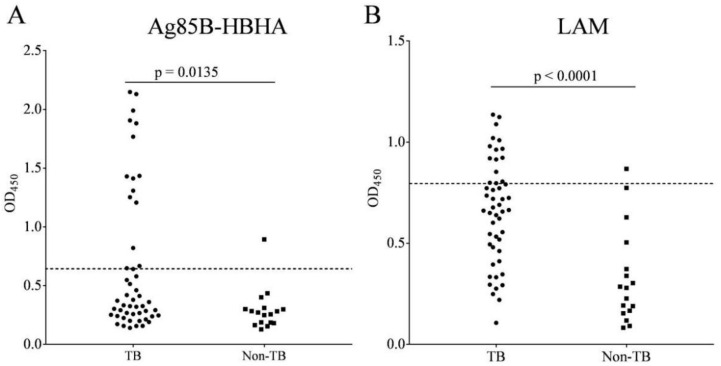
IgG response to Ag85B-HBHA fusion protein and LAM polysaccharide antigen in tuberculosis (TB) patients and non-TB subjects. Scatter plot comparing the IgG levels in response to Ag85B-HBHA (**A**) and LAM (**B**) by ELISA. The graphs show the OD_450_ value of all participants in an ELISA. Horizontal lines indicate the cut-off, based on the mean value from non-TB plus two standard deviations. Differences were assessed by a Mann–Whitney test.

**Figure 4 pathogens-11-01545-f004:**
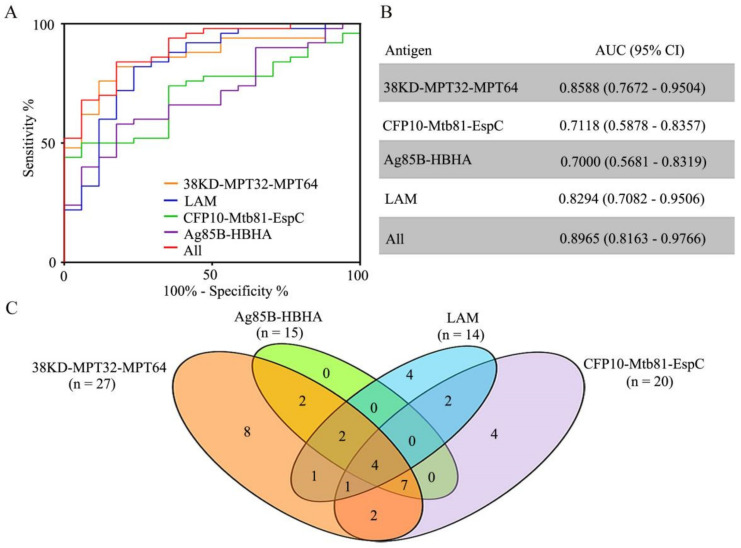
Analysis of the performance of the four antigens in tuberculosis identification. (**A**) ROC analysis was used to evaluate the IgG detection efficiency of the four antigens. (**B**) The area under the curve (AUC) was calculated and values are summarized. (**C**) A Venn diagram of the distributed characteristics of the four antigens in Ab-positive patients (TB˗Ab^+^, n = 37), based on the pooled data presented in Figure 2 and Figure 3.

**Figure 5 pathogens-11-01545-f005:**
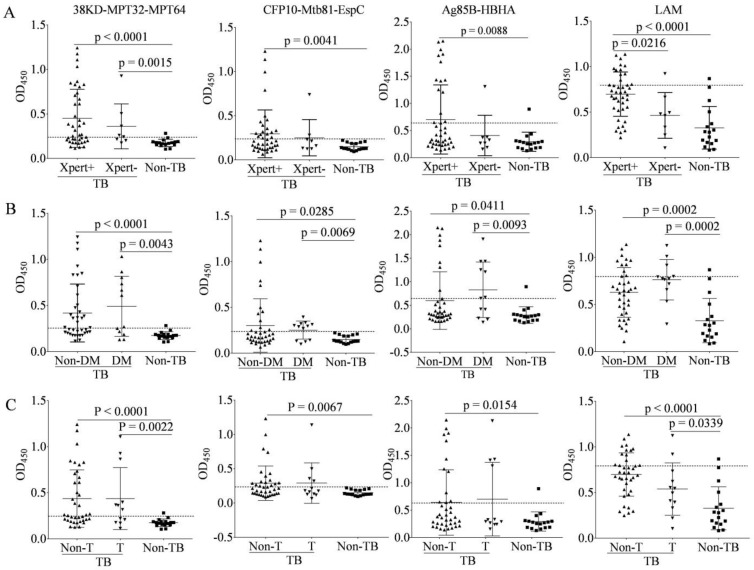
IgG to the fusion protein and LAM polysaccharide antigen in the subgroups of tuberculosis (TB) patients (n = 50) and non-TB subjects (n = 17). Scatter plot comparing the IgG levels in response to the four antigens by ELISA. (**A**) Comparisons among Xpert Mtb-positive TB patients (TB-Xpert^+^, n = 42), Xpert Mtb-negative TB patients (TB-Xpert^−^, n = 8), and the non-TB. (**B**) Comparisons among non-diabetic TB patients (TB-Non-DM, n = 38), diabetic TB patients (TB-DM, n = 12), and the non-TB. (**C**) Comparisons among the non-treatment TB patients (TB-Non-T, n = 38), the initial anti-TB drug treatment group (TB-T, n = 12), and the non-TB. The graphs show the OD_450_ values of all participants based on ELISA. Horizontal lines indicate the cut-off, based on the mean value from the non-TB plus two standard deviations. Differences were assessed by a Mann–Whitney test.

**Table 1 pathogens-11-01545-t001:** Characteristics of the study participants.

Characteristics	PTB (n = 50)	Non-TB (n = 17)
Gender		
Male % (no.)	74.0%(37)	76.5% (13)
Female % (no.)	26.0% (13)	23.5% (4)
Age, mean ± SD, years	45.42 ± 18.59	39.82 ± 5.60
Clinical category		
Mycobacterial culture test		
Culture-positive % (no.)	74.0% (37)	–
Culture-negative % (no.)	26.0% (13)	–
Xpert MTB/RIF assay		
Xpert Mtb positive % (no.)	84.0% (42)	–
Xpert Mtb negative % (no.)	16.0% (8)	–
DM status		
TB-Non-DM % (no.)	76.0% (38)	–
TB-DM % (no.)	24.0% (12)	–
Initial anti-TB treatment status		
TB-Non-T % (no.)	76.0% (38)	–
TB-T % (no.)	24.0% (12)	–

PTB = pulmonary tuberculosis; SD = standard deviation; DM = diabetes mellitus; TB-Non-T = non treatment TB; TB-T = treatment TB.

**Table 2 pathogens-11-01545-t002:** Characteristic of IgG detection to the antigens alone and in combination.

	Sensitivity	Specificity	PPV	NPV	ACC
38KD-MPT32-MPT64	54.0%	94.1%	96.4%	41.0%	64.2%
CFP10-Mtb81-EspC	40.0%	100.0%	100%	36.2%	55.2%
Ag85B-HBHA	30.0%	94.1%	93.8%	31.4%	46.3%
LAM	28.0%	94.1%	93.3%	30.8%	44.4%
38KD-MPT32-MPT64 + CFP10-Mtb81-EspC	66.0%	94.1%	97.1%	48.5%	73.1%
38KD-MPT32-MPT64 + CFP10-Mtb81-EspC + LAM	74.0%	88.2%	94.9%	53.6%	77.6%

Cut-off value were calculated from mean OD_450_ values + 2 SD in non-TB group. PPV = positive predictive value; NPV = negative predictive value; ACC = accuracy.

**Table 3 pathogens-11-01545-t003:** Sensitivities of IgG detection in TB subgroups.

	Xpert MTB		DM Status		Anti-TB Treatment	
	Xpert^+^ (n = 42) %(n)	Xpert^−^ (n = 8) % (n)	TB-Non-DM (n = 38) %(n)	TB-DM (n = 12) %(n)	TB-Non-T (n = 38) %(n)	TB-T (n = 12) %(n)
38KD-MPT32-MPT64	54.8% (23)	50.0% (4)	52.6% (20)	58.3% (7)	52.6% (20)	58.3% (7)
CFP10-Mtb81-EspC	42.9% (18)	25.0% (2)	31.6% (12)	66.7% (8)	42.1% (16)	33.3% (4)
Ag85B-HBHA	33.3%(14)	12.5% (1)	21.1% (8)	58.3% (7)	28.9% (11)	33.3% (4)
LAM	31.0% (13)	12.5% (1)	28.9% (11)	25.0% (3)	31.6% (12)	16.7% (2)
38KD-MPT32-MPT64 + CFP10-Mtb81-EspC	66.7% (28)	62.5% (5)	63.2% (24)	75.0% (9)	65.8% (25)	66.7% (8)
38KD-MPT32-MPT64 + CFP10-Mtb81-EspC + LAM	73.8% (31)	75.0% (6)	71.1% (27)	83.3% (10)	76.3% (29)	66.7% (8)

Cut-off value were calculated from mean OD_450_ + 2 SD in non-TB group. TB-Non-DM = non-diabetic TB; TB-DM = diabetic TB; TB-Non-T = non TB treatment; TB-T = TB treatment.

## Data Availability

The data used to support the findings of this study are included within the article and are available from the corresponding author upon request.

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
