# Peer review of "Enhanced Serum IgG Detection Potential Using 38KD-MPT32-MPT64, CFP10-Mtb81-EspC Fusion Protein and Lipoarabinomannan (LAM) for Human Tuberculosis"

_pathogens, 2022, doi:10.3390/pathogens11121545_

Round 1

Reviewer 1 Report

Comments

The authors constructed and expressed three fusion proteins to facilitate and optimize Mtb-specific Ab detection in PTB patients which includes 38KD-MPT32-MPT64, CFP10- Mtb81-EspC, and Ag85B-HBHA fusion proteins.This manuscript concludes that The combination of 38KD-MPT32-MPT64, CFP10-Mtb81-EspC, and LAM antigens had a relatively high sensitivity (74.0%) and specificity (88.2%) in Mtb-specific IgG antibody detection. The work of this paper is logical. To further improve the manuscript, I have the following suggestions:

Major

1.      Several healthy controls also presented Mtb-specific IgG, would they be latent TB infection?

2.      The sample size of the study is relatively small, especially the health control group.

3.      Add the strength and limitations of the study.

4.      Could the immune response, the sensitivity as well as the specificity of the antigens be impacted by BCG vaccination status?

5.      The TB groups seems older than the HC group.

Minor

1.      Please add more detail information about the study populations, eg., whether some of them are immunosuppressed, diabetes.

2.      It would be better to cite references to explain why specific three proteins were selected for fusion respectively in the introduction section.

3.      Did Ab responses to the four antigens demonstrate different characteristics between patients without anti-TB treatment and 12 patients with anti-TB treatment?Please add the data.

4.      Table 1 is lack of a line in the bottom of the table. Also, the last sentence of the conclusion part has no predicate verb.

5.      The language can be improved by native English speaker.

Author Response

Response to Reviewer 1 Comments

Major

Point 1: Several healthy controls also presented Mtb-specific IgG, would they be latent TB infection?

Response 1: Thank you for the valuable questions. In this study, we mainly focused on TB patients and did not consider latent TB infection indeed. Individuals in the Non-TB group were not excluded by TST or IGRA assays. This is a clear limitation of our study that clarified in the revised manuscript (lines 433-437). Moreover, we have changed “healthy controls (HCs)” to “non-TB subjects (Non-TB)” throughout the revised manuscript. The IgG titer against Mtb-specific antigens in Non-TB seems hardly detected in this study, which indicated that more sensitive detection methods should be used to identify latent TB infection.

Point 2: The sample size of the study is relatively small, especially the health control group.

Response 2: Yes. we carefully considered the problem. The sample size of healthy controls, and patients after grouping was small as well indeed. Supplemental data are difficult to obtain in the short term, because we need to enroll new patients and controls. This study is more inclined to be a preliminary experiment and a polity study. While, our study clear support us to consider more in-depth research, including the possibility of detecting antibodies in various clinical tuberculosis patients. In view of this deficiency, we made a special explanation of this deficiency in the discussion.

Point 3: Add the strength and limitations of the study.

Response 3: Done accordingly (see lines 429-437).

Point 4: Could the immune response, the sensitivity as well as the specificity of the antigens be impacted by BCG vaccination status?

Response 4: In this study, several antigens are coexpressed by BCG and Mtb strains, such as 38KD, MPT32, MPT64, Ag85B, HBHA, and LAM. Theoretically, immunological memory is a remarkable phenomenon after the initial interaction with an immunogen. Memory B cells are a critical reservoir for plasma cell generation in the secondary response. Patients infected with Mtb may have secondary immune response to these antigens on the background of BCG vaccination, which may affects antibody production and even the sensitivity of IgG detection. While, we didn’t find that Mtb-specific IgG detection was impacted by BCG vaccination status. We deleted the corresponding data in the revised manuscript.

Point 5: The TB groups seems older than the HC group.

Response 5: Thanks, this has been further verified. Age showed no statistical significance between TB and HC groups by the Mann–Whitney test (p = 0.2023).

Minor

Point 1: Please add more detail information about the study populations, eg., whether some of them are immunosuppressed, diabetes.

Response 1: Yes, no patients were administered immunosuppressive drugs, and this has been added to the revised manuscript (line 104). Also, there were 38 TB patients with type 2 diabetes mellitus (DM) and 12 patients without DM. Levels of IgG to the three fusion proteins and LAM antigen were compared between TB-Non-DM and TB-DM groups (showed in Figure 5). The detailed results were added in lines 328-331. The IgG levels showed no significant difference between these two subgroups.

Point 2: It would be better to cite references to explain why specific three proteins were selected for fusion respectively in the introduction section.

Response 2: Done, we have added references (16, 17, 18). 

Point 3: Did Ab responses to the four antigens demonstrate different characteristics between patients without anti-TB treatment and 12 patients with anti-TB treatment? Please add the data.

Response 3: We have added these data to Figure 5. Levels of IgG against the three fusion proteins and LAM antigen showed no statistical difference between patients without anti-TB treatment and 12 patients with initial anti-TB treatment. Level of IgG against LAM tended to decrease in the TB-T group compared with the TB-Non-T group, but it didn’t reach statistical significance (p = 0.0541) (see lines 334-338).

Point 4: Table 1 is lack of a line in the bottom of the table. Also, the last sentence of the conclusion part has no predicate verb.

Response 4: Thanks, done accordingly.

Point 5: The language can be improved by native English speaker.

Response 5: The language has been improved by a native English speaker.

Reviewer 2 Report

The number of 50 TB and 17 HC is too small to draw conclusions in this study.

The data is overloaded, the result is not clear.

Positive and negative predictive values should be given. Diagnostic accuracy calculation should be given.

Only patients with culture positivity, which is the gold standard, should be included in this study.

Are there any patients who are both culture and GeneXpert TB negative?

After such intense numerical analysis, tables, figures, diagrams and numbers, I suggest that it be more clear how useful it can be diagnostically in serodiagnosis.

Author Response

Response to Reviewer 2 Comments

Point 1: The number of 50 TB and 17 HC is too small to draw conclusions in this study.

Response 1: Yes. we carefully considered the problem. The sample size of healthy controls, and patients after grouping was less as well indeed. Supplemental data are difficult to obtain in the short term, because we need to enroll new patients and controls. This study is more inclined to be a preliminary experiment and a polity study. While, our study clear support us to consider more in-depth research, including the possibility of detecting antibodies in various clinical tuberculosis patients. In view of this deficiency, we made a special explanation of this deficiency in the discussion.

Point 2: The data is overloaded, the result is not clear.

Response 2: Yes, we have rearranged our results, removed some duplications, and clarified the results (see the revised Ms). 

Point 3: Positive and negative predictive values should be given. Diagnostic accuracy calculation should be given.

Response 3: We have added the positive predictive value, negative predictive value, and accuracy value in Table 2.

Point 4: Only patients with culture positivity, which is the gold standard, should be included in this study.

Response 4: We added and clarified the analysis of positive group of bacteriological examination in revised Ms (lines 322-327).

Point 5: Are there any patients who are both culture and GeneXpert TB negative?

Response 5: There were seven clinically diagnosed TB patients with negative culture and GeneXpert results. IgG detection to 38KD-MPT32-MPT64, CFP10-Mtb81-EspC, and LAM could detect 71.4% (5/7) of these patients as showed in revised Ms (lines 417-420). We have added the diagnostic criteria for tuberculosis referring to WS 288-2017 Tuberculosis Diagnosis published in 2017 [20] (lines 99-100).  

Point 6: After such intense numerical analysis, tables, figures, diagrams and numbers, I suggest that it be more clear how useful it can be diagnostically in serodiagnosis.

Response 6: Done accordingly. See the conclusion in discussion section of revised Ms (lines 429-437).

Reviewer 3 Report

The present manuscript titled: "Enhanced serodiagnostic potential by using ..........and Lipoarabinomannan (LAM) antigen for human tuberculosis" is an attempt to compare and combine TB-antigens-host antibody(response) relationship in the context of PTB diagnosis. They looked for complementarity of the assays proposing auxiliary TB diagnosis. The assays or the antigens are known. However, challenges of serodiagnosis have not been answered. The detection of IgGs (as immune response ) with purified antigen as bait: does not address altered structure or epitopes presented by M.tb within the host. Recent works on LAM structures show that urinary LAM is structurally different than H37Rv-LAM. It also does not evaluate immune-complex (antigen bound antibodies). Overall, high degree of false-positive results make these approaches unattractive. This work needs fresh thoughts and approaches.

Author Response

Response to Reviewer 3 Comments

Point 1: The present manuscript titled: "Enhanced serodiagnostic potential by using ..........and Lipoarabinomannan (LAM) antigen for human tuberculosis" is an attempt to compare and combine TB-antigens-host antibody(response) relationship in the context of PTB diagnosis. They looked for complementarity of the assays proposing auxiliary TB diagnosis. The assays or the antigens are known. However, challenges of serodiagnosis have not been answered. The detection of IgGs (as immune response ) with purified antigen as bait: does not address altered structure or epitopes presented by M.tb within the host. Recent works on LAM structures show that urinary LAM is structurally different than H37Rv-LAM. It also does not evaluate immune-complex (antigen bound antibodies). Overall, high degree of false-positive results make these approaches unattractive. This work needs fresh thoughts and approaches.

Response 1: Thanks for the professional comments, we very much agree with these views. At present, TB antibody is not specific for diagnosis indeed, and can only be used for auxiliary diagnosis cautiously. In this regard, our revision has strengthened that antibody detection is a reference for auxiliary diagnosis of TB. The development direction of antibody detection in TB needs to improve the specificity as much as possible. This article is more about the evaluation of detection sensitivity and specificity, with the purpose of improving the selected antigen combination. A potential use is for the preliminary screening of suspected patients, which is also described in the revised version, and the title has been revised accordingly.

Other relevant issues have been revised accordingly, especially in the discussion sections, including the title of the paper, pointing out possible applications and necessary in-depth research.

Round 2

Reviewer 1 Report

N/A

Author Response

Response to Reviewer 1 Comments

Comments and Suggestions for Authors   N/A

Response: Thank you very much for your professional review work on our manuscript.

Reviewer 3 Report

This is much better written manuscript. However, the basic inadequacy of serodiagnosis made this approach weak. As an attempt towards auxiliary TB diagnosis, this may be acceptable after addition of references. global TB LAM, in-vivo and urinary LAM structures have been elucidated in:

1. ACS Infect Dis 2020, 6, 2, 291-301

2. J Biol Chem 2021, 297, 5, 101265

Author Response

Response to Reviewer 3 Comments

Point 1: This is much better written manuscript. However, the basic inadequacy of serodiagnosis made this approach weak. As an attempt towards auxiliary TB diagnosis, this may be acceptable after addition of references. global TB LAM, in-vivo and urinary LAM structures have been elucidated in:

  1. ACS Infect Dis 2020, 6, 2, 291-301
  2. J Biol Chem 2021, 297, 5, 101265

Response 1: Thank you for your valuable suggestion. We have modified introduction (lines 50-56) and discussion (lines 386-389), added references (5-8 and 31) accordingly, and paid more attention to the current status of TB antibody detection.